# Factors influencing IMF assistance in the Sub-Saharan African region

**Kalindu Abeywickrama[1], Nehan Perera[1], Sithesha Samarathunga[1], Harshani Pabasara[1], Ruwan Jayathilaka[2]\*, Krishantha Wisenthige[1]**

1 SLIIT Business School, Sri Lanka Institute of Information Technology, Malabe, Sri Lanka, 2 Department of Information Management, SLIIT Business School, Sri Lanka Institute of Information Technology, Malabe, Sri Lanka

\* ruwan.j@sliit.lk

**Data Availability Statement:** The data underlying the results presented in the study are available from following IMF UN and The World Bank Databases https://www.imf.org/external/np/fin/tad/

## Abstract

This study examines the determinants influencing the likelihood of Sub-Saharan African (SSA) countries seeking assistance from the International Monetary Fund (IMF). The IMF, as a global institution, aims to promote sustainable growth and prosperity among its member countries by supporting economic strategies that foster financial stability and collaboration in monetary affairs. Utilising panel-probit regression, this study analyses data from thirty-nine SSA countries spanning from 2000 to 2022, focusing on twelve factors: Current Account Balance (CAB), inflation, corruption, General Government Net Lending and Borrowing (GGNLB), General Government Gross Debt (GGGD), Gross Domestic Product Growth (GDPG), United Nations Security Council (UNSC) involvement, regime types (Closed Autocracy, Electoral Democracy, Electoral Autocracy, Liberal Democracy) and China Loan. The results indicate that corruption and GDP growth rate have the most significant influence on the likelihood of SSA countries seeking IMF assistance. Conversely, factors such as CAB, UNSC involvement, LD and inflation show inconsequential effects. Notable, countries like Sudan, Burundi, and Guinea consistently rank high in seeking IMF assistance over various time frames within the observed period. Sudan emerges with a probability of more than 44% in seeking IMF assistance, holding the highest ranking. Study emphasises the importance of understanding SSA region rankings and the variability of variables for policymakers, investors, and international organisations to effectively address economic challenges and provide financial assistance.

## Introduction

The International Monetary Fund (IMF) has been pivotal in global financial stability since its inception at the United Nations Bretton Woods conference in July 1944. It serves as a crisis manager, exerting significant influence over macroeconomic policies and structural reforms in developing nations [1–3]. With 196 member countries, the IMF offers short-term financial loans to nations facing financial crises or liquidity shortages, contingent upon the implementation of economic reforms and structural adjustments [4, 5]. Over its 75-year history, the IMF

extarr2.aspx?memberKey1=895&date1key=2018-09-30 https://www.imf.org/external/datamapper/NGDP https://www.imf.org/external/datamapper/GGXWDG_NGDP@WEO/OEMDC/ADVEC/WEOWORLD https://www.imf.org/external/datamapper/GGXCNL_NGDP@WEO/OEMDC/ADVEC/WEOWORLD https://www.imf.org/external/datamapper/BCA_NGDPD@WEO/OEMDC/ADVEC/WEOWORLD https://www.imf.org/external/datamapper/PCPIEPCH@WEO/OEMDC/ADVEC/WEOWORLD https://data.worldbank.org/indicator/CC.EST https://www.bu.edu/gdp/research/databases/global-china-databases/ https://www.wri.org/update/2022-data-added-china-overseas-finance-inventory https://ourworldindata.org/grapher/political-regime https://www.un.org/securitycouncil/content/countries-elected-members.

**Funding:** The author(s) received no specific funding for this work.

**Competing interests:** The authors have declared that no competing interests exist.

has provided financial and technical assistance through various programmes, including standby agreements and economic growth facilities, with the aim of advising on macroeconomic policies and fostering economic stability [6]. It is essential to note that despite the benefits of seeking IMF assistance, objections can arise regarding the terms of funding, with the pros and cons of the assistance depending on each nation's unique situation.

Nations typically seek IMF assistance due to challenging economic situations such as balance of payments crises, current account imbalances, political inefficiencies, and social vulnerabilities. The likelihood of a nation entering into a fund arrangement within a specific time frame depends on factors unique to each nation, such as past deviations between actual and targeted macroeconomic performance [7]. IMF programmes have mostly benefited low and middle-income countries, positively impacting GDP and government spending, thereby promoting economic growth [8].

Past empirical studies have acknowledged numerous macroeconomic indicators and political variables influencing a country's decision to seek IMF assistance across different time frames and methodologies. Variables such as the reserve position of the country, debt service, real GDP growth, and political and economic factors have been identified as significant determinants [9, 10]. Additionally, the political regime of a nation substantially impacts the decision to seek IMF aid, with democratic regimes participating in IMF programmes in about 15% of all country years and autocracies in approximately 20% of all country years [11]. Scholars have also recognised that alternative financial avenues such as China loans (CL) and associations of the United Nations such as the United Nations Security Council (UNSC) exert an influence over the likelihood of a country seeking IMF aid, which will be comprehensively analysed in the latter parts of this investigation.

By reviewing prior studies, it is evident that most studies measured similar variables such as Gross Domestic Product Growth (GDPG), inflation, exchange rate, balance of payment, and external debt using different periods to determine the factors affecting a particular country's need for IMF assistance. Since the applicability of the research findings in today's context may not provide relevant insights, there is a need to conduct a study to analyse the effects of these variables on seeking IMF assistance in the present context.

The principal contribution of this study is to explore to what extent numerous economic factors proposed in the literature as influencing IMF decisions are determinants of the likelihood that a country signs an adjustment policy with the IMF in Sub-Saharan Africa (SSA). For this purpose, twelve variables have been incorporated into the study, namely, Current Account Balance (CAB), Inflation, Corruption, General Government Net Lending and Borrowing (GGNLB), General Government Gross Debt (GGGD), GDPG, UNSC, Closed Autocracy (CA), Electoral Democracy (ED), Electoral Autocracy (EA), Liberal Democracy (LD), and CL, to determine the likelihood of a SSA country seeking IMF assistance within the time frame of 2000–2022, using a panel probit model. SSA has the second largest and fastest-growing population after the Asian region; ninety percent of SSA countries fall within the lower and lower-middle-income levels facing economic challenges [12]. Notably, thirty of the 36 countries that benefited from the Heavily Indebted Poor Countries (HIPC) initiative started in 1996 to help countries with high debt levels are from African countries [12]. This implies that debt levels held by SSA countries need better management, and policies need implementation to control their excessive debt levels. Moreover, the debt repayment of SSA countries was financed by the structural adjustment facility established to fund low-income developing countries [13]. Selecting SSA for this study is a prudent choice as it is positioned prominently on the IMF commitment list due to its high indebtedness, and its repeated reliance on loans makes it a compelling subject for analysing the country's economic conditions and policy implications.

The present study differs from existing research and fills a gap in the literature in many ways. Firstly, the study employs the latest data collected from 2000–2022, making it unique in the coverage of the years. Secondly, the findings of this study emphasise the predominant variables influencing SSA countries to seek IMF assistance, examining the consequences of the CL on the likelihood of seeking IMF aid, providing valuable insights for policymakers. Thirdly, very few studies have used the panel probit methodology incorporated in this study, which offers more insights about the included variables. Moreover, the hypothesis style used in this study's results is a novel concept that has not been frequently used in prior literature. Additionally, a limited number of studies have assessed the impact of the political regime of a country on seeking IMF assistance along with macroeconomic indicators. Simultaneously, the present research fills the gap in existing knowledge by encompassing all the macroeconomic indicators and regime types that were studied separately in previous literature.

Furthermore, this study conducted extensive analysis using predicted probability to rank countries in SSA accordingly to determine which countries had been highly influenced by macroeconomic indicators. The results of this study emphasise the ranking of highly vulnerable countries in SSA in seeking IMF assistance. The insights concluded that Sudan, Burundi, and Guinea-Bissau are the first three primary nations in SSA seeking IMF assistance in all three-time frames. This study holds significant importance for SSA countries by offering valuable insights in numerous ways. Furthermore, this academic work goes beyond previous studies by using predicted probability to rank SSA countries based on their vulnerability to seeking IMF assistance. This helps policymakers strategically direct their efforts and resource allocation for SSA countries facing critical conditions. Moreover, the knowledge provided is crucial for understanding the economic vulnerabilities in SSA and thereby formulating improved fiscal policies and structural reforms for better external shock management, fostering economic stability, and promoting sustainable growth in SSA countries.

The subsequent sections of this research paper will present a literature review comprising similar past studies, data and methodology, results, and discussion, and finally provide a conclusion and recommendations for policy implications.

## Literature review

Prior studies, have examined various macroeconomic and political factors influencing a country's probability of seeking IMF assistance across various nations, using a variety of macroeconomic and political indicators. This study conducts a literature review using twelve variables: CAB, GDPG, inflation, corruption, GGGD, GGNLB, UNSC, CA, ED, EA, LD, and CL, explicitly focusing on the SSA region to examine historical findings regarding how these variables affect a country's tendency to seek IMF support.

The relationship between the IMF and SSA has been multifaceted, shaped by historical trends and evolving economic conditions. The formation of the extended fund facility in 1974 permitted short-term financial assistance, adapting to participant countries' needs. SSA nations, mainly employing mechanisms like the extended fund facility and trust funds, have been involved with the IMF to address numerous economic challenges. Historically, in 1945, Ethiopia marked the region's first membership, and most nations joined in the early 1960s. In the 1970s, SSA countries faced economic challenges due to internal and external shocks, leading them to seek IMF assistance [13, 14]. In the 1980s, several SSA nations turned to the IMF for funding through structural adjustment programmes to address economic recessions. Subsequently, SSA nations sought support through poverty reduction strategy programs, revealing the IMF's influence on healthcare in Ghana, Uganda, and Tanzania and, in the case of Zimbabwe in 1990, seeking IMF support under global scrutiny [13, 15, 16]. This climaxes the ongoing

difficulties in the IMF and SSA relationship, demanding a nuanced understanding and frequent assessment of the fund's role in fostering sustainable economic development in the region.

The study analyses the critical role of macroeconomic policies in affecting employment consequences amid output growth in the SSA region. From 1991 to 2016, across thirty-seven countries, outcomes discovered that policy effectiveness is improved through adaptable and targeted tax frameworks, rigid labour markets, and liberalised trade. Notably, investment and industrial policies demonstrate nuanced influences, affected by changes in government sizes and foreign investment proportions [17]. This underscores the relationship between macroeconomic policy adjustments and their effectiveness as labour market adjustment tools in the SSA region, especially preferring more formal employment groups.

SSA wrestles with a challenging capital squeeze worsened by global inflation, severe fiscal policies, rising borrowing costs, and exchange rate pressures. The surge in inflation and public debt, driven by market-based subsidies and weakening aid, reveals macroeconomic imbalances, excessively affecting vulnerable populations. Simultaneously, SSA faces an insistent capital crisis within mounting macroeconomic imbalances fuelled by the Ukraine war and COVID-19. The region's vulnerability demands vital policy involvement and international assistance to steer these economic challenges. Inflation continues at high levels, doubling since the pandemic, and the reduction in fuel prices joined with assistance from phasing out countries like Senegal and Ethiopia, amplifies inflation volatility. The Gambia, Mali, Rwanda, and Cameroon wrestle with rising public wages, living costs, and fuel prices [3, 12, 18, 19]. The IMF plays a vital role in the SSA region, supporting robust debt resolution, international support, and policy involvement. The IMF's supervision is essential in directing the multifaceted economic landscape and offering solutions to achieve stability. The economic challenges and macroeconomic disparities intensified by COVID-19 and geopolitical actions like the Ukraine war pose weighty threats to the region. Adequate policy arrangements and international funding are imperative for SSA to navigate these economic challenges, highlighting the urgency for sustainable debt resolution and aid mechanisms.

The IMF's crucial role in SSA cannot be exaggerated. The IMF provides essential guidance for sustainable solutions as the region wrestles with financial challenges and economic imbalances. This initial literature highlights the significance of the IMF in the SSA region, highlighting the complex relationship between the IMF and SSA. The following literature will further classify and discover these dynamics based on the variables in the study.

## Current account balance

The CAB is a critical component of a country's balance of payments, replicating its external trade performance and can be in deficit or surplus. CAB is a crucial gauge of economic health, influencing foreign investors' confidence and potentially leading to a current account reversal. The CAB directly mirrors savings-investment linkages and fiscal factors, providing insights into an economy's progress and operations in the global market [20].

CAB deficits, exceeding five per cent in many African countries, raise concerns about debt accumulation and sustainability. In the mid-1990s, despite positive macroeconomic growth in SSA, persistent CAB imbalances raised alarms about sustainability and the danger of economic crises. They revealed that countries like Zambia, Mozambique, Mali, Lesotho, Gambia, and Seychelles exhibit sustainable deficits. At the same time, Rwanda, Burundi, Togo, and Burkina Faso face unsustainable ones [21]. The study underscores the importance of examining the CAB in the SSA region, shedding light on economic challenges allied with prolonged deficits and their probable impact on long-term fiscal stability. The trade imbalances arise as a primary

contributor, highlighting the critical need to inspect CAB in the SSA region. Addressing economic challenges associated with enduring deficits is vital for long-term stability, requiring effective policies.

The study analyses the severe role of CAB in the SSA region, revealing that trade liberation has led to increased imports surpassing exports. This imbalance poses a significant danger to economic growth, as increasing CAB deficits may obstruct growth efforts [22]. This awareness is critical for policymakers in SSA, highlighting the need for strategic involvement to achieve current account dynamics in pursuit of sustained economic development in the region. This study has yet to consider the full range of factors prompting the SSA region, such as regional conflicts and non-tariff barriers.

A study employing panel regression and desiring a fixed effect model examines the interplay of net foreign assets, savings, and CAB in the SSA region from 1980 to 2013, showing positive influences of savings and net foreign assets on the CAB, while population growth, dependency ratio, and foreign direct investment exhibit adverse effects [23]. The study mentions policies prompting net foreign assets and savings. In contrast, effective management of foreign direct investments, highlighting initiatives to boost exports, and inspiring domestic savings while addressing demographic challenges become vital for improving CAB in the region.

In Nigeria, CAB, comprising goods, services, and income, mirrors diverse movements influenced by international oil market dynamics: surpluses during high prices and deficits during low prices. The study employed macroeconomic indicators and the intertemporal model, showing that throughout periods of excessive CAB deficit, the Nigerian economy obeyed the intertemporal budget constraints despite macroeconomic instability [24]. The node between fiscal and CAB deficits was discovered through a study revealing a significant influence of CAB deficits on Nigeria's fiscal balance due to heavy dependence on foreign exchange from oil proceeds [20]. A study employed a panel autoregressive distributed lag model for Western African states, emphasising the optimistic impact of GDP on the long-run CAB [25]. An investigation was carried out to inquire into the determinants of CAB in Ghana, Cote d'Ivoire, and Nigeria, utilising the value-at-risk approach, emphasising real income as significant in explaining the CAB [21]. Investigations into CAB determinants in Cote d'Ivoire, Ghana, and Nigeria stress the critical importance of understanding CAB dynamics in the SSA region.

The CAB is a vital gauge of a country's economic health, reflecting its external trade performance and prompting foreign investor confidence. Understanding the CAB's role in SSA is critical for policymakers, highlighting the need for strategic involvement to attain balanced CAB dynamics and foster sustained economic expansion in the face of global market dynamics and regional complications.

## Corruption

Corruption is a strategy typically used to influence people away from the proper path of action or duty of conduct, either in executing their official tasks or in activities related to political or economic issues [26]. Furthermore, corruption is simply denoted as a private gain at public expense [27]. It arises when the expected benefits exceed the predicted cost, and the benefits are not limited to financial gains but consist of acquiring political office, power, and prestige [28]. Corruption has numerous economic consequences on the allocation of resources, on the way economic decision-makers will analyse different actions, on GDP growth, and also negatively affects the level of trust between people, which will endanger the stability of social and political institutions [29]. Corruption is a global phenomenon that impacts both developed and developing nations.

Corruption is a prominent feature of African politics, with numerous high-profile scandals [30, 31]. For example, Nigeria's Sani Abacha [32] and South Africa's Jackie Selebi [33] are identified as public officials involved in major corruption scandals. Moreover, corruption in Africa is still widespread and remains among the world's most severe, with low average incomes, low literacy rates, and various authoritarian governments [34–37]. According to the data taken from Transparency International, six African countries are rated as highly corrupt, another 35 countries are considered very corrupt, and no African country is included among the least corrupt group [38]. Evidence from prior studies proves that countries such as the Democratic Republic of Congo, Nigeria, Uganda, Ethiopia, and Kenya have been identified as nations with highly destabilised governments, which led to inadequate provision of services such as electricity and fuel [39]. Under the situation under the corrupt and autocratic president Jose Eduardo dos Santos, who has governed for 35 years, billions of dollars have flowed to a small elite as kids starve in Angola, which has the highest mortality rate, emphasising the corruption in the African region more [40].

Furthermore, Eritrea is identified as corrupted to a high degree as there is severely inadequate funding for public bureaucracies [41, 42]. It is a visible fact that development in the African context is exceptionally vulnerable to corruption. In contrast, African politics frequently exhibits a high degree of patrimonial politics and bureaucratic forms of domination [43, 44]. In addition, the educated youth, who had earlier been identified as more likely to bring corruption to people's attention, was a major driving force behind Egypt's protests against daily lives marked by corruption at all levels [38]. Another damaging form of corruption can be stated as misusing the funds allocated for health and other social services, and undoubtedly, the lives of Africans are negatively affected by such issues [30, 31].

Findings of a report convey that the funds received to oppose the 2014 Ebola outbreak needed to be more documented, and appropriately, one-third of the funds have been misused [45].

## Gross Domestic Product Growth

The GDPG rate, measured as the annual percentage variation in a country's GDP, is a dynamic gauge of economic well-being. It indicates the proportion of growth or contraction in the value of goods and services produced within a country. Policymakers use the GDPG rate to indicate economic performance, enabling changes to monetary policies [46]. Understanding the GDPG is essential for policymakers' economists and investors, as it provides insights into a country's overall economic performance and trends.

SSA's economic performance reaped consideration due to its deterioration in the final half of the 20[th] century. Optimism in the 1960s was replaced by decelerated growth and contraction in the 1980s, leaving the region's living standards behind. Political reasons point to poor strategies hindering growth, while exogenic factors consider external features affecting African economies. The COVID-19 pandemic has strictly impacted SSA development, with a sharp deterioration predicted from 2019 to 2020. This economic turmoil threatens to untie years of expansion gains in the region, worsened by SSA countries' high requirement for external funding, and another study conducted research into the dual influence of dropping oil prices and COVID-19 on Africa's GDPG. Oil-dependent economies suffer the most with a -10.75% growth loss [47–50]. The COVID-19 pandemic has resulted in a significant economic collapse. The region's weakness towards outside funding further complicates regaining efforts.

SSA's complex economic growth is hampered by worsening inequality, persistent poverty, and rising unemployment. Due to low GDP per capita in many SSA member countries, living standards have remained relatively high. A significant issue is youth unemployment, which is

correlated with low productivity. Another study highlights the inconsistency of economic growth in SSA, driven by demographic dynamics and the imperative for effective policies to connect the region's demographic advantage for sustainable economic growth [51, 52]. This underscores the crucial need for policies addressing the complex challenges hindering SSA's pursuit of sustainable economic growth. The effort on unemployment growth and determined challenges explores extremely precise policy interferences. It highlights the necessity for practical strategies, investments, and formal environments to connect the SSA region's demographic advantage for sustainable economic growth.

The study groups countries in terms of GDPG and poverty reduction. Nations facing hasty GDPG, but limited poverty reduction include Mozambique, Nigeria, Zambia, and Burkina Faso. In the case of Tanzania, high population growth, slow household consumption, and overestimated economic growth contribute to tepid poverty reduction. In Zambia, increased poverty and negative growth are experiential. In the case of Nigeria, stagnation in living standards despite economic growth is positive in terms of trade. Another cluster, categorised by economic deterioration and political instability, comprises Madagascar and Cote d'Ivoire. Kenya and Cameroon show unstable growth records with partial long-term gains in the welfare gauges [53]. This study emphasises the multifaceted connection between GDPG and poverty consequences across diverse SSA regions. In developing nations, the international financial crisis significantly impacted the GDPG, with an alteration of more than 8% between pre-crisis and the newest IMF forecast from 2008–2011 [54]. The study reveals that countries with lower pre-crisis GPDG rates were likely to participate in IMF programs. Moreover, poor countries received more funds, aligning with the present literature on IMF performance. This emphasises the importance of GDPG rates manipulating countries' engagement with IMF programmes during economic crises.

## General Government Gross Debt

GGGD refers to the short- and long-term loans obtained by governments to fund public deficits incurred due to the higher spending programme than the anticipated revenue. Further debt can be incurred and acquired both domestically and overseas. Many economies borrowed domestically or externally to fund the budget deficit to respond to global economic downturns. According to the World Bank, countries experiencing a rapid rise in debt show frequent fragility linked to various reasons, such as weak governance, and empirical evidence depicts that it is highly improbable for a nation to run a surplus budget. As a result, public debt accumulation is unavoidable [46, 55]. In order to understand a nation's economic situation and comprehend its ability to fulfil its financial commitments, it is imperative to monitor GGGD, and policymakers should work to maintain sustainable debt levels and the requirement for public spending.

According to prior studies, a high value of GGGD is significant in SSA countries as a result of both internal and external shocks [56]. The escalating debt accumulation became unsustainable, resulting in repayment difficulties and a debt crisis in the 1990s, reducing growth and other development objectives. The revival of GGGD in SSA countries can be attributed to factors such as the 2008 global financial crisis shocks, adverse commodity price shocks, financial gaps in infrastructure, and a drop in official development assistance. The outbreak of COVID-19 was the most significant cause for the increase in government spending, which led to a rise in debt profiles to address financial shortfalls [57, 58]. Studies show that out of 36 countries that have benefited from the HIPC initiative introduced by the World Bank and IMF, 33 countries are associated with SSA, which suggests debt levels are poorly managed, necessitating measures to reduce excessive debt levels. Consequently, in 2005, the multilateral debt relief

initiative offered US$ 76 billion in debt-service relief to 36 countries, of which 30 are in Africa [56, 59].

Further, in 2020, the IMF's executive board approved a loan worth US $4.3 billion in its Radio France Internationale scheme for SSA countries to mitigate the economic and social effects of the pandemic [60]. Despite the IMF and World Bank's intervention through the HPIC project to mitigate the debt burden of most SSA countries, some continued to have a significant level of debt as of 2015 [59]. For example, the high public debt stock in 2000 caused Ghana to be designated as HPIC in 2001. By 2008, Ghana shifted from a low-income HPIC country to a lower middle-income economy, with a reduced public debt of 182% in 2000 to 32% in 2008 [46]. However, the debt level in Ghana started rising again, from 26.2% of GDP in 2006 to 57.2% of GDP in 2016, with GGGD reaching 63% of GDP as of December 2019. Additionally, Nigeria experienced an increase in foreign debt service as a percentage of exports, rising from 12.7% in 2013 to 45.7% in 2016 [59, 61]. The trends discussed highlight the continued difficulties in managing and sustaining debt levels in these countries, emphasising the necessity for efficient debt management strategies.

In addition, several countries in the SSA, such as 29 years for the Congo, 27 years for Gambia, 24 years for Zambia, 21 years for Madagascar and Tongo, 18 years for Mali, 17 years for Burundi, and Equatorial Guinea, have been in high debt regimes [62]. When considering the period 2012 to 2017, Equatorial Guinea, the Congo Republic, Ethiopia, Zambia, and Cameroon have been stated as the countries with the fastest rise in GGGD, and concurrently, the debt ratio remained low in most of the SSA countries after the HIPC debt relief [59].

An extensive amount of empirical literature exists on the effects of the debt threshold on economic growth. Still, the few reviewed literature in SSA depict mixed results regarding this matter. For example, Ghana had a bidirectional relationship with positive effects on public debt and GDP, while Nigeria reflected a weak association [63]. After investigating the association between external borrowings and GDP in SSA, 31.3% debt-GDP was recommended for the country [64]. However, the ratio between government borrowings and GDP increased to 59.3% in 2019, and economic activities slowed to 0.7% in 2019 compared to the 4.2% statistic in 2000 [60].

Further studies imply that in April 2005, the IMF and World Bank developed the dent sustainability analysis to assist countries in financing development needs [59]. These findings suggest that seeking IMF assistance helps a country curtail GGGD and regain economic stability.

## General Government Net Lending and Borrowing

GGNLB represents the deviation between a government's total revenue and expenditure over a specified period. It plays a crucial role in a government's fiscal policy and is frequently used to evaluate the government's financial position. Furthermore, this can be depicted in terms of surplus or deficit, signifying when the government's total revenue exceeds expenditure or vice versa. Many governments in the nation face tremendous pressure due to increased public expenditure, which encompasses both capital and recurrent expenses, coupled with the challenges of low revenue generation.

SSA, the second-largest region with the fastest-growing population, is predicted to witness a significant increase in demand for essential services, including healthcare, education, housing, and infrastructure, as its working population is expected to rise from 705 million in 2018 to nearly 1 billion by 2030 [65, 66]. In 2018, Nigeria, Ethiopia, Egypt, the Democratic Republic of the Congo, Tanzania, South Africa, Kenya, Uganda, Algeria, and Sudan were considered the top ten most populous countries in SSA, which put the African government under great stress due to increased expenditure compared to revenue generation [67]. When assessing the

economies from 1980 to 2012, per capita income, tax share, minimum wage, population growth, foreign aid, public debt, and democracy, along with oil revenue, GDP population, trade openness, oil price, taxation and inflation, are highlighted as crucial factors influencing the growth of government expenditure both in Ghana and Nigeria [68]. In addition, the gains in export revenue in SSA are often leveraged by the additional borrowings in countries such as Ghana, Kenya, and Madagascar, which led to a considerable increase in current and capital expenditures [69]. Further, all the above incidents emphasise how crucial debt management is to mitigate potential risk.

Most countries in the SSA suffer from budget deficits where fiscal policy reforms are essential to raise public revenue and to restructure public expenditure allocation to elevate economic activity [70]. Under this condition, SSA countries experienced international public financial management standards with the assistance of the International Development Association, the World Bank, and the IMF, and thereby, in 2007, the Central African Republic (CAR) was accepted by the IMF as having reached HPIC status. The republic was closely administered to manage debt and control expenditure [65]. Moreover, public procurement is a crucial economic activity regarding public expenditure. The term comprises functions such as buying, renting, and acquiring supplies related to obtaining goods and services by the government to serve its citizens [71]. To ensure the value of money, maintain sanity, and enhance public financial management within the Ghanaian public sector, the Public Procurement Act was enacted in 2003. This act regulates all the procurement activities of the government. After recognising certain shortcomings in the country's public procurement system, such as a lack of a comprehensive strategy to coordinate and specify general procedures, the government decided this act was necessary [72].

Moreover, in Ethiopia, the public procurement of goods and services accounts for over 60% of the total government expenditure. The HPIC initiative stipulated various reforms in the country's public financial management, resulting in a US$ 1275 million aid package with 47.2% relief on external debt with the assistance of the International Development Association [46, 73]. The findings suggest that implementing reforms to manage government expenditures with IMF assistance has helped the SSA countries regain economic stability.

## Inflation

Inflation, defined as the rate of rise in prices over a specific time frame, serves as a comprehensive measure of a country's overall price and cost of living fluctuations. It can be calculated broadly for goods or services. Inflation measures the level to which goods and services become more expensive over a period, typically a year. This highlights the consequences of inflation as a vital economic gauge, reflecting the dynamic fluctuations in pricing and cost structures within an economy [74].

In the 1980s, fiscal policies in the SSA prioritised monetary deficits, leading to overvalued exchange rates and high inflation. Policymakers in the region continue to wrestle with the critical challenge of controlling inflation pressures, where the instability in headline inflation is heavily influenced by unstable agricultural production, impacting the SSA's consumer price index [75, 76]. Handling inflationary pressures is an endless challenge for policymakers, exacerbated by fluctuating food prices resulting from unpredictable agricultural output. Addressing these problems is crucial for bringing about economic stability in the area.

Inflation in SSA poses challenges to the finance and growth relationship. The study highlights a severe inflation threshold of 31%, beyond which financial development inversely affects economic growth. Holding inflation below this threshold is critical for nurturing positive financial growth dynamics. SSA faces the challenge of dealing with inflation to foster a

healthy financial growth link and safeguard sustainable development [77]. The discoveries are specific to only twenty-three selected SSA region countries and may only be appropriate to some regions due to varied economic structures and conditions.

In Nigeria, policymakers face a significant challenge in addressing inflation, exacerbated by the need to stimulate domestic demand after an economic recession. Since 1970, price levels have shown significant variations, highlighting the complexity of maintaining stability. Meanwhile, in South Africa, oil prices have a complex relationship with inflation, indirectly influencing refined products consumer prices indirectly through agriculture. The rise in oil values highlights the complex interaction of inflation dynamics in the SSA [78, 79]. In Nigeria, determined challenges with inflations highlight the complications of handling domestic demand and fiscal obligations following economic fluctuations. In South Africa, the complex connection between oil prices and inflation dynamics highlights the interconnected nature of these issues, specifically in agriculture.

In the SSA region, inflation poses momentous challenges to economic stability. The analysis's emphasis on Ghana and South Africa emphasises the complexity of the inflation and growth relationship. High inflation levels have adverse effects, indicating potential economic vulnerabilities [80]. Challengers in dealing with inflation may influence the monetary policy outline in the SSA, demanding tailored policies to safeguard sustainable economic growth within fluctuating inflationary burdens. The study emphasises Ghana and South Africa, and its discoveries may not be easily inferred to other countries in SSA due to heterogeneity in policies, economic structures, and external factors.

In SSA, inflation poses a significant economic risk, and high inflation compromises stability, hence necessitating focused policies for growth and resilience. Factors like oil prices intricately affect inflation, which climaxes the requirement for tailored policies.

## China loan

China's lending to foreign nations has emerged as a cornerstone of its global economic engagement, prominently demonstrated by initiatives such as the Belt and Road Initiatives (BRI). Utilising a combination of bilateral agreements, concessional funding, and commercial loans, China has extended significant financial support to countries spanning Asia, Africa, Latin America, and beyond. Notably, China provided aid totalling US$ 354.4 billion to other nations from 2000 to 2014, with official creditors including the central government of China, government agencies such as the Ministry of Commerce, and lending by state-owned policy banks like the China Development Bank and the China Export-Import Bank. China currently stands as the foremost provider of foreign aid, independent of the Organisation for Economic Cooperation and Development Assistance Committee [81, 82]. These approaches underscore China's ambition to foster economic development, yet they also raise concerns regarding the terms of these loans and their long-term sustainability.

The escalating public debt in SSA nations has ignited global discussions regarding sustainability within the region, leading to evolving relationship between China and SSA countries. Over the past two decades, China's economic engagement with SSA nations has shown a consistent rise since the inception of the Forum on China-Africa Cooperation in 2000 and the establishment of the China-Africa Development Fund in 2006. China can be considered Africa's main bilateral partner in various socioeconomic development projects across the continent, facilitated by initiatives like BRI and coordination with the African Union, while offering loans exceeding $150 billion and deepening economic ties between the two regions [83]. The pressing demands for infrastructure and economic development in SSA countries have increased their reliance on China for financial assistance, resulting in China's massive loans to

the SSA region surpassing those of other countries. From 2000 to 2020, China provided 1188 loan agreements totalling $160 billion to numerous African governments, with primary beneficiaries including countries such as Angola, Ethiopia, Zambia, Nigeria, the Republic of Congo, and Ghana. Chinese loans in Africa have primarily focused on the transport, power, and mining sectors, while also playing a critical role in African resource extraction operations, including mineral oils and gas, contributing to the expansion of these resource allocation sectors to promote economic growth [84, 85]. In navigating the evolving relationship with China, SSA nations are seeking to strike a balance between protecting their economic autonomy and leveraging the benefits of Chinese investments, for sustainable development outcomes.

Chinese lending has emerged as a notable alternative to traditional sources of finance such as the IMF, but past studies predict that China will struggle to exert influence over the IMF due to its weakness at the time of the institution's founding [86]. Observers began to recognise the potential for Chinese loans to weaken the IMF's bargaining position in negotiations as early as 2004 when China joined bailout negotiations with the IMF in countries such as Pakistan. Additionally, countries that don't benefit from Western favouritism at the fund can refine their bargaining position by turning to China [87]. Moreover, China's ability to establish its initiatives has rivalled that of the World Bank. Some evidence suggests that these efforts have been successful, as countries less favoured by Bretton Wood institutions have shown high-level support for these Chinese-led organisations [88]. The decision of the Angolan government in 2004 to seek assistance from Chinese loans after the civil war by negotiating oil-backed loans with the China Export and Import Bank to reconstruct infrastructure, and the leader of Hezbollah stating that China could help Lebanon rather than the IMF to recover from the economic crisis in August 2020, are further incidents illustrating a trend where countries increasingly consider China as a viable alternative to the IMF for financial assistance and economic reform [89, 90]. However, the status of the IMF as the most senior lender of sovereign debt regimes makes establishing a rival to the IMF far more difficult. Nonetheless, these incidents highlight a striking trend where countries increasingly view China as a viable alternative to the IMF for financial assistance and economic reform.

## United Nations Security Council

The UNSC stands as one of the six principal organs of the United Nations entrusted with preserving global peace and security. Comprising 15 members, it includes 5 permanent members with veto power, namely, the United States, Russia, the United Kingdom, China, and France, alongside 10 non-permanent members appointed by the UN General Assembly for 2-year terms. The UNSC possesses the authority to issue resolutions that bind member states, with powers encompassing the determination of peacekeeping strategies, imposition of international sanctions, and authorisation of military action [91, 92].

The relationship between the UNSC and the SSA region has been intricate. The UNSC plays a pivotal role in addressing conflicts and fostering peace and security in the region, often through the deployment of peace operations. While many of these operations have been successful in saving lives and mitigating violence, some countries have faced criticism for their limitations and shortcomings, including issues related to the comprehensiveness peace agreements, and a military-centric focus. Additionally, structural and procedural challenges within the UNSC, such as disagreements among members and closed-door decision-making, have at times impeded effective action in response to crises in the SSA region. Moreover, emerging challenges such as climate change are increasingly acknowledged as significant factors influencing security and development in the region [93–95]. This underscores the necessity for collaboration between the UNSC and the SSA region to effectively address these complex issues.

The relationship between the UNSC, the IMF, and the SSA region is multifaceted, with implications for global finance, geopolitics, and growth assistance. The study examines the significant effects of IMF lending conditions. Nations serving on the UNSC face about 30% fewer conditions on their IMF loans, indicating a potential trade-off between geopolitical considerations and conditionalities. Furthermore, the geopolitical impact extends beyond IMF lending to financial market repercussions and economic stability. Nations entering IMF programs during their UNSC tenure experience higher bond and bill yields, currency depreciation, and weaker stock market performance. This suggests reduced confidence in the nation's fiscal assets due to perceptions of political favouritism influencing IMF programs. Additionally, the influence of short-term political motivations, such as UNSC membership, extends to the effectiveness of foreign aid in the SSA region. Aid disbursed during a nation's UNSC tenure demonstrates significantly lower effects on economic growth, indicating that short-term political favouritism may diminish the effectiveness of aid in achieving development outcomes [96–98]. This highlights the complexity of geopolitics, such as UNSC membership, which can substantially affect IMF lending practices and the efficacy of foreign aid, with implications for economic stability and growth in the region.

## Regime

The term "regime" can denote a form of government or a system of management, encompassing the ruling administration of a nation or a method of governance. Its connotations vary depending on whether it is used in a political, social, or organisational context [99].

In this study, political regimes are classified into four distinct types: CA, EA, ED, and LD [100]. CA is characterized by limited political freedoms and stringent government control over the political landscape, including media and opposition activities. Globally, there is an upward trend of autocratic tendencies eroding democratic principles and adopting authoritarian governance structures in many nations [101, 102]. EA represents a hybrid regime, blending democratic procedures with authoritarian methods. While elections occur, they often fall short of democratic standards, with the dominant government exerting influence over the process and civil society, curtailing media freedom and opposition activities [103]. ED enables citizens to elect representatives through free and fair elections, featuring consistent multiparty polls with secret ballot voting. It upholds civil liberties such as freedom of expression, association, and the press, ensuring a robust framework for political representation beyond electoral processes [104]. LD constrains government power and safeguards individual freedoms through established standards and institutions, prioritising norms like pluralism, civil rights maintenance, and dispute resolution within a constitutional framework. Its framework encompasses elections, the rule of law, equal protection of rights, a market economy, and separation of powers. The sustainability of these regimes hinges on their steadfast commitment to democratic principles, governance, and human rights globally.

The correlation between regime characteristics and economic growth in the SSA region has been the subject of numerous studies employing varied methodologies. One study, utilising the general method of moments, examined the influence of remittances and regime durability on economic growth across 33 SSA nations from 1970 to 2012. It found that while remittances did not consistently affect economic growth, regime durability exhibited a negative relationship with growth, while regime type showed a positive association. Another analysis employed a panel vector autoregressive model within a general method of moments framework to analyse the link between energy consumption, economic growth, and democracy in 16 SSA nations from 1971 to 2013. The interaction variable of energy consumption and democracy was found to positively influence economic growth, emphasising the role of democratic

governance in promoting the energy consumption-growth nexus [105–107]. This underscores the intricate relationship between regime characteristics and economic growth in the SSA region.

In the study analysing political regimes and IMF programme participation, it was found that regime types significantly influence nations' decisions to engage with the IMF. Party-based autocracies balance sovereignty effects and programme benefits, especially during challenging economic crises. Personalist regimes prioritise economic benefits over sovereignty costs, typically participating in IMF programmes only during economic recessions. In contrast, military regimes exhibit indifference to both political costs and economic crises, diverging from other autocratic responses [11]. This nuanced understanding of regime influences sheds light on why certain nations join IMF programmes while others refrain, indicating regime-specific rationales and restrictions.

Comprehending the intricate dynamics between political regimes and economic outcomes is crucial for formulating effective policies and fostering sustainable development. As research delves deeper into the complexities of regime influences, it underscores the need for nuanced governance approaches and international cooperation in addressing the challenges and opportunities of the modern world.

## Theoretical framework

In order to address the identified gaps in this analysis, a conceptual framework has been developed to evaluate the probability of macroeconomic and political indicators influencing the decisions of countries within the SSA region to seek assistance from the IMF. Fig 1 presents the conceptual framework devised for this study to assess the likelihood of macroeconomic and political indicators affecting countries' decisions within the SSA region to seek support from the IMF.

A conceptual framework crafted to explore the relationship between political and macroeconomic indicators and the decisions of nations within the SSA region to seek assistance from the IMF. The framework incorporates the dependent variable "Seeking IMF assistance" and encompasses five macroeconomic independent variables: GDP growth (GDPG), General Government Net Lending and Borrowing (GGNLB), General Government Gross Debt (GGGD), inflation, Current Account Balance (CAB), along with corruption, hybrid regimes

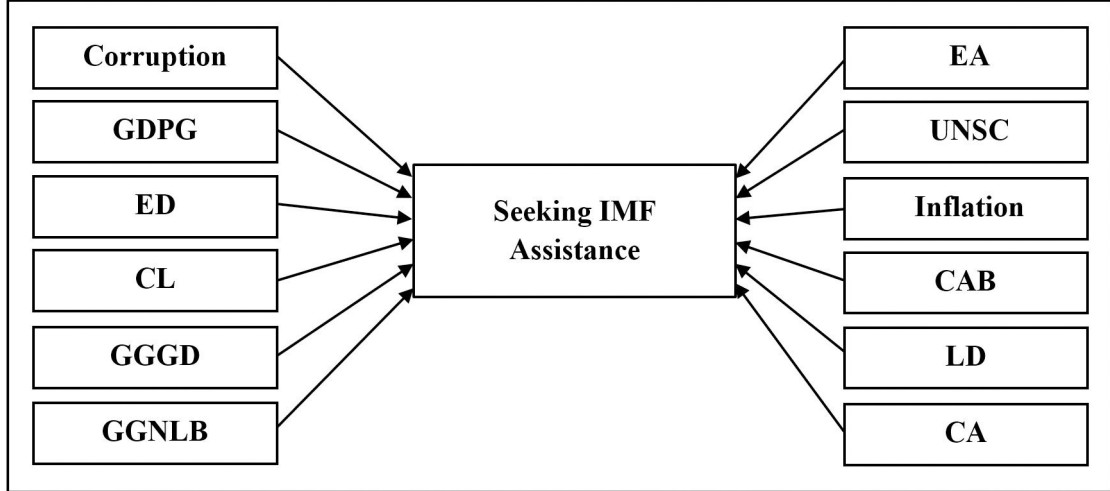

**Fig 1. Conceptual framework.** Source: Authors' illustration.

(EA), membership in the United Nations Security Council (UNSC), liberal democracies (LD), electoral democracies (ED), and closed autocracies (CA) as six political independent variables, and China's lending (CL) as the availability of external options for financial assistance as an independent variable. This framework aims to shed light on how various macroeconomic and political factors influence countries' decisions to seek support from the IMF within the SSA region.

## Data and methodology

This section outlines the data sources and statistical model employed in this study.

### The data

For this study, a comprehensive dataset covering 160 countries worldwide was identified. Although the study aimed to compile evidence for all 196 countries globally, data was accessible for 160 countries. Specially focusing on the SSA region, the study observed 45 countries within the region. However, data could only be sourced from 39 countries. The data file used for the study is presented in the S1 Appendix. Data acquisition involved meticulous exploration of information from reliable sources, notably the IMF database, the World Bank, and Our World in Data spanning 2022. Table 1 provides details of the data sources and variables.

### The model

This study seeks to estimate the likelihood that determinants influencing the decision to seek IMF assistance related to macroeconomic and political variables. The data analysis conducted using a panel probit model, and the stepwise method was employed to establish the final model identification. The general specification for the panel probit model is given by:

$y_{it}$ = 1 if a country is seeking IMF assistance.

$y_{it}$ = 0 if a country not seeking IMF assistance.

$$(Iy = 1|x_{it}, c_i) = \Phi(x_{it}\beta + c_i) \tag{1}$$

$i = 1,\ldots,n$ and $t = 1,\ldots,T$, The panel probit model is estimated using the maximum likelihood form of the Eq (1). The result is present in Eq (2). When $y$ is the observed outcome, $\Phi$ represents the individual-specific effect [50]. The dependent variable is a panel dummy variable, taking the value of 1 if a country is seeking IMF assistance and 0 otherwise if a country is not seeking IMF assistance. Thus, the model to be estimated is

$$P(IMF = 1)$$
$$= \Phi(\beta_0 + \beta_1 cab + \beta_2 ggnlb + -\beta_3 gggd + \beta_4 inflation + \beta_5 corruption + \beta_6 gdpg + \beta_7 CL$$
$$+ \beta_8 LD + \beta_9 ED + \beta_{10} EA + \beta_{11} UNSC + \beta_{12} CA + c_{it} \tag{2}$$

Where:
- The IMF dummy variable was directly obtained from the IMF official website
- CAB, GGNLB, and GGGD are expressed as a percentage of GDP
- GDPG is expressed as an annual percentage change
- The variable inflation measures the annual percentage change in end-of-period consumer prices
- The variable corruption measures the control of corruption estimate
- Regime types (CA, EA, ED, and LD) are represented as dummy variables
- UNSC and CL are included as dummy variables

**Table 1. Data sources and variables.**

| Variable | Definition | Measurement | Source |
|---|---|---|---|
| **Dependent Variable** | | | |
| | Seeking IMF assistance | | IMF Database https://www.imf.org/external/np/fin/tad/extarr2.aspx?memberKey1=895&date1key=2018-09-30 |
| **Independent Variables** | | | |
| GDPG | Gross Domestic Product Growth | Annual percentage change % | IMF Database https://www.imf.org/external/datamapper/NGDP |
| GGGD | General Government Gross Debt | Percent of GDP | IMF Database https://www.imf.org/external/datamapper/GGXWDG_NGDP@WEO/OEMDC/ADVEC/WEOWORLD |
| GGNLB | General Government Net Lending and Borrowing | Percent of GDP | IMF Database https://www.imf.org/external/datamapper/GGXCNL_NGDP@WEO/OEMDC/ADVEC/WEOWORLD |
| CAB | Current Account Balance | Percent of GDP | IMF Database https://www.imf.org/external/datamapper/BCA_NGDPD@WEO/OEMDC/ADVEC/WEOWORLD |
| Inflation | Inflation | End of the period Consumer prices | IMF Database https://www.imf.org/external/datamapper/PCPIEPCH@WEO/OEMDC/ADVEC/WEOWORLD |
| Corruption | Corruption | Control of corruption; Estimate | The World Bank https://data.worldbank.org/indicator/CC.EST |
| CL | Chian Loan | 1 if China Loan | https://www.bu.edu/gdp/research/databases/global-china-databases/ https://www.wri.org/update/2022-data-added-china-overseas-finance-inventory |
| ED | Elector Democracy | 1 if Elector Demography | https://ourworldindata.org/grapher/political-regime |
| LD | Liberal Democracy | 1 if Liberal Democracy | |
| EA | Elector Autocracy | 1 if Elector Autocracy | |
| CA | Closed Autocracy | 1 if Closed Autocracy | |
| UNSC | United Nations Security Council | 1 if UNSC | https://www.un.org/securitycouncil/content/countries-elected-members |

The estimation process involves step-by-step procedures using different estimators. Following econometric tests assessing the significance of both individual-specific effects and the sample average for covariates, a correlated random effects probit model is preferred. Maximum likelihood, the random effects probit model is preferred in this procedure, as a fixed effect is not feasible [108].

## Hypotheses and results

This section summarises twelve hypotheses and results that affect a country's decision to seek assistance from the IMF. In the analysis, the descriptive statistics in S2 Appendix expose wide variations across SSA regions in key macroeconomic and political indicators, influencing their probability of seeking IMF support. For instance, the highest mean and standard deviation across all variables and countries are for the inflation variable in Sudan.

Table 2 presents panel probit model estimated results with marginal effects. The estimated coefficients, standard errors, and marginal effects expressed in percentages are reported for each variable included in the model.

The analysis examines hypotheses regarding the likelihood of a country seeking IMF support, analysing the impact of macroeconomic and political variables. The hypotheses are empirically tested using a panel probit model, with results detailed in Table 2. To get more insight into the marginal effects, S3 Appendix presents a panel probit regression analysis of step-by-step results examining the determinants of seeking IMF assistance in the SSA region.

**Table 2. Panel probit model estimated results with marginal effect.**

| Variables | Estimate | Standard Error | Marginal Effect (in percentage) |
|---|---|---|---|
| Constant | -2.299 | 0.29582 | |
| Corruption | -0.441[a] | 0.07791 | -44.1[a] |
| GDPG | -0.037[a] | 0.00663 | -3.7[a] |
| ED | 0.598[a] | 0.15623 | 59.8[a] |
| CL | -0.212[a] | 0.07461 | -21.2[a] |
| Ln GGGD | 0.150[b] | 0.06132 | 15[b] |
| GGNLB | -0.016[b] | 0.00799 | -1.6[b] |
| EA | 0.290[c] | 0.15072 | 29[c] |
| UNSC | -0.202 | 0.14020 | -20.2 |
| Inflation | 0.003 | 0.00229 | 0.3 |
| CAB | -0.004 | 0.00418 | -0.4 |
| LD | 0.197 | 0.20841 | 19.7 |

Note: The coefficients presented above represent marginal effect with the significant levels as follows, a significant at 1%, b at 5%, and c significant 10% significance level.

In the study, S4 Appendix provides valuable insights into the macroeconomic indicators, showcasing line graphs demonstrating fluctuations for each variable across the nations in the SSA region. The variable CAB reveals substantial variations in nations such as Chad, CAR, Mozambique, and the Seychelles within the timeframe of 2000–2022. Notably, these fluctuations reveal dynamic changes in economic conditions within the SSA region over the years. Moreover, the variable corruption reveals noteworthy patterns in the SSA region, where many nations demonstrate negative values, showing a high level of corruption. Conversely, nations like the Seychelles and Botswana in the region exhibit positive values, implying less influence from corruption. This underlines the varying degrees of corruption prevalence across diverse nations within the SSA region, exposing the complex socio-economic landscapes present in the SSA region.

Incorporating an approach inspired by influential research, the study presents results in a hypothesis style, inspecting the likelihood of countries seeking IMF support through macroeconomic and political variables [109, 110]. The study introduces novel hypotheses, and all the hypotheses are summarised as follows:

**Hypothesis 1: $\beta_1 > 0$**
The likelihood of a country seeking IMF assistance decreases as corruption increases.
**Hypothesis 2: $\beta_2 > 0$**
The likelihood of a country seeking IMF assistance increases as GDPG decreases.
**Hypothesis 3: $\beta_3 > 0$**
The likelihood of a country seeking IMF assistance increases as ED increases.
**Hypothesis 4: $\beta_4 > 0$**
The likelihood of a country seeking IMF assistance increases as CL in a country decrease.
**Hypothesis 5: $\beta_5 > 0$**
The likelihood of a country seeking IMF assistance increases as the GGGD in a country increase.
**Hypothesis 6: $\beta_6 > 0$**
The likelihood of a country seeking IMF assistance increases as GGNLB decreases.
**Hypothesis 7: $\beta_7 > 0$**
The likelihood of a country seeking IMF assistance increases as EA increases.

**Hypothesis 8: $\beta_8 > 0$**

The likelihood of a country seeking IMF assistance increases as UNSC membership decreases.

**Hypothesis 9: $\beta_9 > 0$**

The likelihood of a country seeking IMF assistance increases as Inflation increases.

**Hypothesis 10: $\beta_{10} > 0$**

The likelihood of a country seeking IMF assistance increases as CAB decreases.

**Hypothesis 11: $\beta_{11} > 0$**

The likelihood of a country seeking IMF assistance increases as LD increases.

**Hypothesis 12: $\beta_{11} > 0$**

The likelihood of a country seeking IMF assistance increases as CA increases.

Hypothesis 1 suggests that the probability of a country seeking IMF assistance from the IMF is estimated to show a negative correlation with the level of corruption. This variable's expected sign is positive. The result shows that when corruption increases, a country seeking IMF assistance will decrease. The marginal effect of corruption reveals that a one per cent increase in the control of corruption estimated in the leads to an increase in the probability of a country seeking IMF assistance by 44.1%. All other factors are constant. The study's variable corruption is statistically significant at the 1% significance level. The empirical findings denied the expected positive relationship that suggests an expansion of corruption will raise the probability of a country pursuing IMF assistance. Unflatteringly, when the level of corruption increased, the IMF hesitated to release funds. This was proven by a statement from IMF spokesperson Gerry Rice, highlighting that the outlook of the international community influences the IMF's decisions. Mr. Rice noticed a lack of clarity within the global community concerning the appreciation of the government in Afghanistan, where Afghanistan could not access IMF resources or special drawing rights [111]. In an extensive analysis discovering the determinates of decisions on IMF credit, which employed a panel model using 118 countries spanning the period from 1971 to 2000, it was found that countries with a more unbalanced and polarised political system face increased challenges in framing a credible adjustment programme, compelling them to seek support from the IMF [9]. In conclusion, the study challenges the estimated positive link between corruption and looking for IMF support, revealing a shocking negative correlation. This denies expectations and aligns with the IMF's cautious response to increased corruption, as seen in Afghanistan.

Hypothesis 2 shows an inverse relationship between a country seeking IMF assistance and GDPG. When GDPG increases, correspondingly, a country seeking IMF assistance will decrease. The marginal effect of the GDPG reveals that the probability of seeking IMF assistance increase by 3.7% when the GDPG as an annual percentage change decrease by one per cent. All other factors constant. The study, which analyses whether one model will fit all to analyse the determinants of IMF arrangements, used the variable GDPG. Their results also proved a negative sign for GDPG, demonstrating a meaningful relationship between weak economic growth and the likelihood of countries seeking IMF assistance [112]. The analysis supports hypothesis 2, revealing that as GDPG declines, the probability of a country seeking IMF assistance increases. The study highlights the vital role of economic stability in influencing countries' decisions to request support from the IMF.

Hypothesis 3 examines the probability of a country seeking IMF assistance from the IMF, which is estimated to show a positive correlation between ED and IMF. Thus, the expected sign of the ED variable is positive. The marginal effect of the ED shows that the probability of a country seeking IMF assistance is increased by 59.8% when the dummy is equal to 1 when all other factors are held constant. The variable ED is statistically significant at a 1% significance level. Analysing data from 1970 to 2006, the study shows that during moderate economic

recessions, ED is less likely to seek IMF support. Conversely, during severe economic crises, EDs are more likely to seek IMF support than autocracies. This trend is recognised in democratic leaders' electoral weakness, which improves the political motivation to focus economic cries through IMF programmes [113]. These patterns increase as elections approach, emphasising the relationship between domestic politics, economic conditions, and regime types in shaping nations' decisions concerning IMF programme participation.

The explanation for hypothesis 4 is that the probability of a country seeking IMF assistance from the IMF is estimated to show an inverse relationship with CL. This correlation indicates that when CL increases, the probability of a country seeking IMF assistance decreases. The marginal effect of the CL shows that the probability of a country seeking IMF assistance is increased by 21.2% when the dummy is equal to 1 when all other factors are held constant. The variable CL is significant at a 1% significance level. Analysis of data from 2000 to 2018 reveals that nations facing defaults on Chinese debt seek IMF support during rigorous adverse shocks. IMF programmes often impose severe loan conditions, altering governance and potentially leading to earlier exits of corrupt leaders. China's lending is often coupled with non-cash repayment or geopolitical concessions, and sometimes it alternates for IMF support. However, IMF participation weakens political benefits for leaders of indebted nations as it requires increased fiscal transparency, risking exposure to corrupt practices [86, 114, 115]. China has emerged as a crucial player in the SSA region's economic development, spending $ 160 billion across 1,188 loan contracts. China has formed itself as the SSA region's leading bilateral partner, supporting fundamental financial support for infrastructure and economic development programmes [85]. The dynamic relationship between CL, IMF support, and governance structures in the SSA region highlights the complex challenges and opportunities facing indebted nations. While Chinese funding has been influential in supporting economic development, IMF involvement can decline political advantages for leaders and impose greater financial transparency. As the SSA region maintains its ability to steer these dynamics, policymakers must thoroughly balance the benefits and risks correlated with numerous sources of financial support to ensure sustainable growth and stability.

Hypothesis 5 examines the probability of a country seeking IMF assistance from the IMF, which is estimated to show a positive correlation between GGGD and IMF. Thus, the expected sign of the GGGD variable is positive. The GGGD variable in the comprehensive model unanticipatedly showed a negative sign, which contradicted the estimated positive correlation. A focused simple regression was conducted, closely fitting the expected positive sign. A logarithmic transformation was applied to the GGGD variable, effectively aligning it with the intended positive relationship. The marginal effect of the GGGD shows that the probability of a country seeking an IMF is increased by 15% when the log of GDP change as a percentage rise by 1% when all other factors are held on constant. The variable GGGD is statistically significant at a 5% significance level. In line with the study's conclusions on the threshold effects of public debt on economic growth in Africa, in 1996, various African nations received debt relief through the World Bank and IMF's HIPC initiative [62]. The primary objective of the HIPC initiative was to support countries burdened with high debt levels by helping reduce debt [59]. Limited relative studies on African public gaining reforms and financial management expose a case of the CAR and highlights the significance of financial assistance and debt relief [65]. The findings echo the importance of debt relief programmes, as proved by Africa's prior experiences with the HIPC programme.

The explanation for the hypothesis 6 is that the probability of a country seeking IMF assistance from the IMF is estimated to show an inverse relationship with GGNLB. This variable's expected sign is negative. This correlation indicates that when GGNLB increases, the probability of a country seeking IMF assistance decreases. The marginal effects show that GGNLB

increased by 1% as a percentage of GDP, estimated to decrease the likelihood of a country seeking IMF assistance by 1.6% when all other factors held constant. The variable GGNLB is significant at a 5% significance level, and a study of limited relative studies on African public gaining reforms and financial management exposes the importance of public expenditure accountability. Examining the case of CAR emphasises the importance of continuous monitoring for effective public financial management [65]. In conclusion, the findings highlight the relationship between a GGNLB and a country's likelihood of seeking IMF assistance and highlight how critical responsible public expenditure and financial management are in influencing a country's economy.

Hypothesis 7 examines the probability of a country seeking IMF assistance from the IMF, which is estimated to show a positive correlation between EA and IMF. Thus, the expected sign of the EA variable is positive. The marginal effect of the EA shows that the probability of a country seeking IMF assistance is increased by 29% when the dummy is equal to 1 when all other factors are held constant. The variable EA is statistically significant at a 10% significance level. The relationship between EA and an enhanced tendency to seek IMF assistance. This tendency is recognised as the comparative vulnerability of economic growth under autocratic regimes versus democracies. Output downfalls are found to be more prevalent under autocracy, potentially leading such regimes to pursue external financial assistance, incorporating from the IMF, to steer economic challenges. Furthermore, autocratic leaders are more tender to the political costs associated with seeking IMF support, observing it to control crises while easing domestic political influences. Subsequently, the IMF may assist as a rational option for autocratic regimes to address economic difficulties while managing potential political fallout [11, 116].

Hypothesis 8 examines the UNSC membership effect of a country seeking IMF assistance. The expected sign of this variable is negative. The marginal effects of the UNSC show that the probability of a country seeking IMF assistance increases by 20.2% when the dummy is equal to 1 when all other factors are held constant. In the analysis, the variable UNSC membership was found to be statistically insignificant. This suggests that membership in the UNSC does not have a significant likelihood of seeking IMF assistance. The study reveals that non-permanent members of the UNSC are more likely to receive IMF loans compared to other nations. Additionally, being part of the UNSC appears to reduce the number of conditions imposed on IMF programmes [117, 118]. This highlights a nuanced correlation between temporary membership in the UNSC and obtaining support from the IMF. This underscores the complicated relationship between UNSC membership and the decision to seek IMF assistance.

The logic behind hypothesis 9 is that there is a positive correlation between a country seeking IMF assistance and inflation. When inflation increases, correspondingly, a country seeking IMF assistance will increase. This study is intended to investigate whether there is any positive impact on a country seeking IMF assistance due to inflation. The marginal effect of inflation shows that the probability of a country seeking IMF assistance increases by 0.3% when the inflation adjusts annual percentages of end-of-period consumer prices year on year. Change is increased by 1%, while all the other factors held constant. In the analysis, inflation emerges as statistically insignificant at a 10% significant level; also, in a study of factors influencing IMF approval of financial arrangements, the variables exhibit an unexpected sign and are insignificant [119]. This underscores the complex relationship between inflation and IMF approval. The extreme bounds analysis discovers issues influencing IMF credit decisions and underscores that countries with high inflation are more likely to require IMF assistance, but the IMF may be less eager to offer funds in such cases, leading to inconsistent negative results [120]. The analysis shows inflation to be statistically insignificant in seeking IMF support. This aligns with a broader study, emphasising the complex link between inflation and IMF approval. The

study underscores the complex mechanisms that lead to complicated and varied results, where high inflation makes regions more inclined to seek IMF assistance. However, the IMF reveals caution in granting aid in such situations, resulting in complicated and varied outcomes.

Hypothesis 10 examines the CAB effect of a country seeking IMF assistance. The CAB variable harms a country seeking IMF assistance. The expected sign of this variable is negative. However, the finding shows that CAB is an insignificant relationship between countries seeking IMF assistance. The probability of a country seeking IMF assistance increases by 0.4% when the percentage of GDP of CAB decreases by 1%. All other factors held on constant. The anticipated sign for CAB was a negative relationship, and the results align with this expectation.

Nevertheless, the variable CAB did not reach statistical significance at the 10% significance level. A study using a pooled sample of annual observations for 91 emerging countries also noted a negative sign for the coefficient. CAB was not statistically significant at the 15% significance level on the decisions related to IMF assistance, warranting further investigations [119]. An analysis was conducted to evaluate the likelihood of a country devaluing its currency within the context of an IMF programme. The study focused on 48 developing countries during the Bretton Woods era. The findings suggest that the extent of CAB worsening is less significant in influencing a country's decision to seek IMF assistance [121]. Despite the forecast negative relationship, the results underscore no substantial change in a country seeking IMF assistance. This underscores the complexity of CAB's influence, supported by the analysis of developing countries and historical analysis, highlighting the need for more investigation in understanding the dynamics of IMF-related decisions.

Hypothesis 11 examines the LD effect of a country seeking IMF assistance. The expected sign of this variable is positive. The finding shows that LD has an insignificant relationship between countries seeking IMF assistance. The marginal effect of the LD shows that the probability of a country seeking IMF assistance increases by 19.7% when the dummy is equal to 1 when all other factors are held constant. The study analyses that LD nations expose sustained growth associated with autocracies, with output collapses being more enduring under autocratic rule. Furthermore, LD reveals greater resilience, easing the severity of crises and preventing persistent periods of autocracy [122]. This highlights a weakened need for IMF involvement in LD, as they are better equipped to control shocks and maintain stability autonomously.

These results impact valuable insights into the discourse, paving the way for more informed policy approaches in the context of global financial support.

## Discussion

The discussion explores the role of macroeconomic and political indicators in determining eligibility for IMF assistance. Using panel probit modelling, the study analysed the predicted probabilities for each country across three-time frames: 2000–2022, 2017–2022, and 2019–2022. Fig 2, presented in correlation with S5 Appendix, illustrates the probability levels of a country seeking IMF support from 2019 through 2022. The colour variety is revealing, with the darkest blue representing countries with the highest possibility, ranging from 50% to 40%, and the lightest blue representing the least likelihood, ranging from less than 10% of seeking IMF assistance. Black signifies the countries omitted from the study within the SSA region. Sudan is in the darkest blue grouping, signifying the highest likelihood of seeking IMF support. Meanwhile, Eswatini, Seychelles, Botswana, Rwanda, and Benin fall into the lightest blue region, demonstrating the slightest possibility of seeking IMF aid.

The analysis emphasises a steady trend across three time-based segments: 2000–2022, 2017–2022, and 2019–2022. Sudan, Burundi, and Guinea-Bissau emerge as the primary nations seeking IMF assistance in all three timeframes.

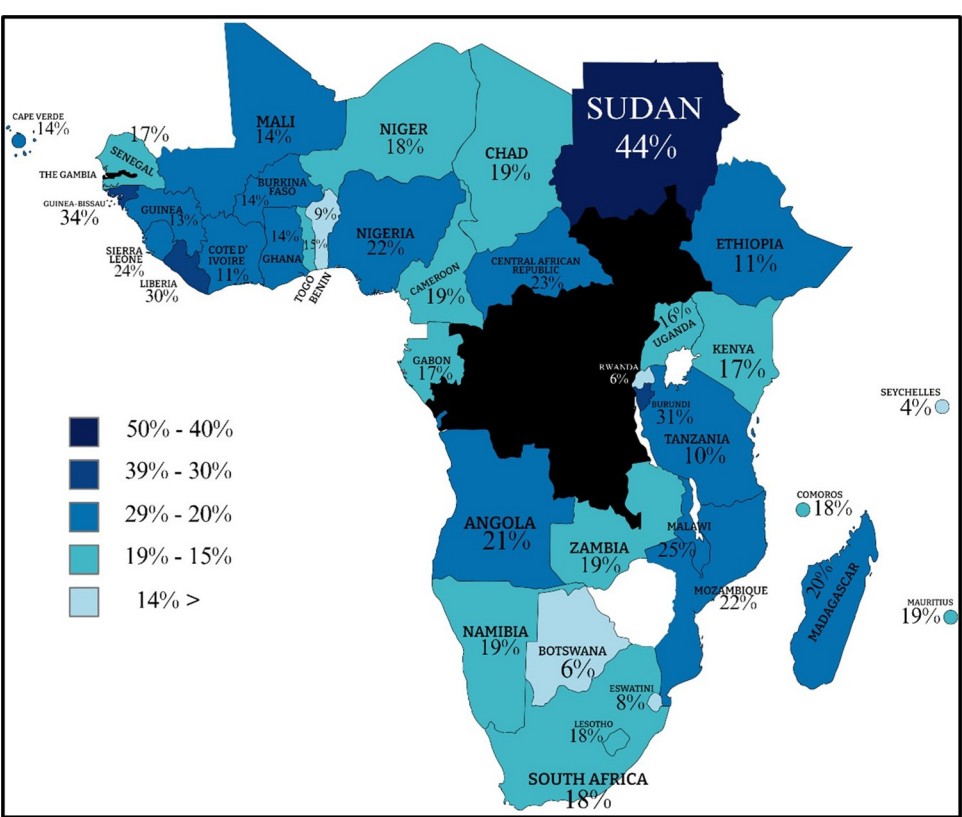

**Fig 2. SSA countries predicted probability percentage of seeking IMF assistance from 2019–2022.** Reprinted from
https://mapchart.net.under a CC BY license, with permission from Minas Giannekas, original copyright 2024. Source:
Authors' illustration based on predicted probabilities.

The statistics shown in S5 Appendix emphasise the vital insights of the SSA region in seeking IMF assistance. According to the projecting analysis, Sudan claims the highest spot in seeking IMF assistance in all three sections, with the likelihood of seeking aid from the IMF exceeding 42% in the 2017–2022 and 2019–2022 segments, while it exceeds 26% in the 2000–2022 segment analysis. Sudan's debt is composed of long-standing debts, posing weighty financial challenges. Sudan's economy in 2020, contracted by -3.6% due to COVID-19 and flooding challenges. The pandemic worsened transitional government hurdles, but the international community, eased by the IMF and World Bank, assisted Sudan's debt [123]. This highlights Sudan's significant challenges and the nation's economic struggles, worsened by COVID-19, which are being addressed with international support.

In the analysis, Guinea-Bissau secured the second spot, and Burundi claimed the third spot in seeking IMF assistance. In all three segments, Guinea-Bissau maintains a likelihood of over 20%, while Burundi exhibits over 22% probability from 2000–2022, 2017–2022, and 2019–2022, increasing to 28%, emphasising an intensified likelihood of seeking IMF support. Burundi shows economic persistence while dealing with COVID-19 and the consequences of the Ukraine war. By 2022, the IMF expects a robust growth rate of 3%, but ongoing macroeconomic issues such as poor terms of trade and domestic inflation threaten living standards [124]. Burundi and Guinea-Bissau face economic challenges, and Burundi's resilience within COVID-19 and the Ukraine war is notable.

In the 2000–2022 analysis, Liberia claims the second spot, but in 2017–2022 and 2019–2022, Liberia consistently holds the fourth position. In the 2000–2022 and 2017–2022 analyses,

Sierra Leone secured the fifth position. During 2014–2016, Sierra Leone was impacted by the Ebola crisis, prompting significant international and private financial support [125]. This shows the growing economic challenges and the fluctuating landscape engagement within the SSA region. Sierra Leone's historical significance under colonial rule as a naval base, commercial hub, and centre of education contrasts sharply with its post-independence marginalisation. Lacking the charismatic leadership of figures like Nkrumah or Nyerere, it hasn't reached global fame. The 1997 military involvement, the article explores, stemmed from historical factors involving economic challenges like deteriorating terms of trade and inconsistent development policies worsened by IMF and World Bank programmes [126, 127]. These initiatives influenced a marginalised class of intellectuals unable to find employment, exhibiting Sierra Leone's broader political and economic underdevelopment.

Angola, CAR, and Chad consistently rank in the top ten countries seeking IMF assistance, with a probability surpassing 14%. In 2020, the COVID-19 pandemic impacted Chad's economic recovery, contracting GDPG by 0.9%, driven by challenges in agriculture and oil and worsening fiscal and CAB conditions. The CAR depends on timber and diamonds but was formerly considered the world's poorest country and faces economic hurdles due to weak governance and regional conflicts where the CAR grapples with arrears to Angola and is negotiating with Angola for a credible plan [128, 129]. These findings focus on the critical roles of countries in the SSA region in seeking IMF support and unravelling complex economic challenges essential for regional stability and global economic dynamics.

Nigeria consistently secured the eighth spot in the 2017–2022 and 2019–2022 analyses, with an over 21% chance of seeking IMF support. The study analyses the economic complications of numerous political regimes in Nigeria from 1984 to 2015. It analyses the impact of conflict and corruption on economic development indicators, specifically democracy versus dictatorship. Results discover that democracy, devoid of conflict and corruption, fosters long-term economic growth, while autocracy delays it. Corruption appears to be a substantial problem, diminishing long-term economic progress. The effect of conflict on economic development remains ambiguous [130, 131]. The findings highlight the essential role of robust anti-corruption procedures and conflict-resolution mechanisms in Nigeria's democratisation process, which are necessary for maintaining economic advancement.

In the comprehensive study of the dataset, Sudan appears as the country with the highest chance and ranks as the first country to seek IMF assistance. However, the actual engagement happened only once in 2021. Notably, Sudan witnessed a reduction in corruption scores during this period. Contrastingly, low-income countries like Malawi and Sierra Leone have sought IMF support eight times. In comparison, Niger and Madagascar did so seven times, and Chad and Burkina Faso demonstrated significant engagement, seeking IMF support six times within the 2000–2022 timeframes.

In the lower levels of ranking, prominent variations in IMF assistance likelihood are detected across all three sections in the SSA region. Botswana maintains the lowest probability throughout the 2000–2022 analysis, while Seychelles claims the slightest likelihood of seeking IMF assistance in the 2017–2022 and 2019–2022 analyses. Nevertheless, Seychelles experiences variations, securing the thirty-sixth and thirty-ninth spots in 2000–2022 and 2019–2022. Botswana defies the resource curse narrative with its notable economic growth, mainly attributed to abundant diamond resources. Botswana has the most outstanding economic performance in the SSA region [132]. Despite being a high-income, tiny island state, Seychelles has required IMF support numerous times from 2000 to 2022. This inconsistent situation is driven by specific challenges, including the effects of the COVID-19 crisis and the vital need to preserve macroeconomic sustainability. Seychelles has the highest GDPG in the SSA region [133].

This emphasises that even wealthy states can face economic complications, prompting engagement with the IMF for critical assistance.

Rwanda consistently ranks low in all three segments seeking IMF support, with less than a 6% likelihood. Namibia, Cabo Verde, Sao Tome and Principe, and Eswatini show fluctuating probabilities and surpass the 10% mark at specific times. In 2019–2022 and 2017–2022, Benin's likelihood of seeing IMF assistance is less than 11% and over 11% in 2000–2022. In a study investigating Cabo Verde, Eswatini, Sao Tome and Principe, and Seychelles, they shared small populations and faced economic volatility due to factors like natural disasters and limited diversification. Programme engagement in these countries, including stand-by arrangements, extended fund facilities, and policy support instruments by the IMF, has produced satisfaction. Emergency funding throughout COVID-19 was timely. Despite heightened programme engagement, only Comoros hosts a resident IMF representative office [134]. The discoveries expose the changing probabilities of seeking IMF support among SSA regions. Regions like Rwanda consistently show low likelihoods, while other nations, such as Namibia and Cabo Verde, experience variations. The study on Eswatini, Sao Tome and Principe, Cabo Verde, and Seychelles emphasises their unique challenges and satisfaction with IMF programmes during the COVID-19 pandemic.

This discussion provides valuable insights into the SSA region rankings and the likelihood of seeking IMF assistance across different segments. This discussion is crucial for policy-makers, investors, and international organisations to recognise economic weaknesses, tailor financial aid, and design targeted involvement.

## Conclusion

This study explores the likelihood of a country seeking IMF assistance in SSA nations, employing the data collected over 23 years. The consequences of the study point out that corruption and GDPG exert the most substantial impact on SSA countries seeking IMF assistance, while CAB, inflation, LD, and UNSC demonstrate an inconclusive outcome in SSA countries seeking IMF assistance. At the same time, the political regime of CA is omitted from this study. Moreover, in contrast to the prevailing notions that corruption prompts a nation to seek IMF assistance, this study fundamentally highlights corruption as a pivotal macroeconomic indicator that limits a country's tendency to seek IMF assistance. Further implications of the study suggest that Sudan, Burundi, Liberia, and Guinea-Bissau emerged as the primary nations to request IMF aid within the time scope, and it also identified Seychelles as a high-income country, having sought IMF assistance more than six times during the period.

### Policy implications

The findings of this study provide significant implications for policymakers in the SSA region. For instance, policymakers should prioritise concentrating on anti-corruption measures and governance reforms to mitigate high corruption levels, and it is also implied that prioritising investment in infrastructure, education, and technology is paramount to withstand economic challenges and promote sustainable economic growth. Further, the study highlights that effective debt management approaches and tailored financial support are essential for combating the burden of high government debt and financial imbalances in SSA nations. Additionally, studying and understanding how the political regimes of a country influence seeking IMF aid helps policymaker tailor policy responses to the ongoing regime. For instance, it is notable that nations with autocratic regimes are more likely to seek IMF assistance as a result of a lack of transparency, accountability, and effective decision-making. There, the policymakers will be able to prioritise implementing government reforms aimed at solving those issues to promote

economic stability. Ultimately, the focus should be redirected towards formulating comprehensive fiscal and monetary policies to address structural issues contributing to economic vulnerabilities and reduce the SSA countries' reliance on external financial assistance.

## Limitations of the study

Despite the findings of this study being credible, it is important to admit that there are limitations as well. The unavailability of data about the incorporated variables for some SSA nations constrains this study. Hence, the study excluded four SSA countries among the forty-five nations in SSA.

## Future research

The future study endeavours could extend the model by considering the influence of the selected variables on the SSA countries that were excluded from the current analysis, and this expansion would provide a more inclusive perspective across a broader range of SSA nations. Additionally, the scope of the study could be broadened to encompass other regions and their influences on seeking IMF assistance.

## Supporting information

**S1 Appendix. Data file.**
(XLSX)

**S2 Appendix. Summary descriptive statistics of each country.**
(DOCX)

**S3 Appendix. Panel probit estimated results and marginal effect in percentage.**
(DOCX)

**S4 Appendix. Line graph analysis of macroeconomic variables across the SSA region.**
(DOCX)

**S5 Appendix. SSA countries predicted probability ranking of seeking IMF assistance.**
(DOCX)

## Author Contributions

**Conceptualization:** Kalindu Abeywickrama, Nehan Perera, Harshani Pabasara, Ruwan Jayathilaka, Krishantha Wisenthige.

**Data curation:** Kalindu Abeywickrama, Nehan Perera, Harshani Pabasara.

**Formal analysis:** Kalindu Abeywickrama, Nehan Perera, Sithesha Samarathunga, Harshani Pabasara, Ruwan Jayathilaka.

**Investigation:** Kalindu Abeywickrama.

**Methodology:** Kalindu Abeywickrama, Nehan Perera, Ruwan Jayathilaka.

**Software:** Kalindu Abeywickrama, Nehan Perera, Harshani Pabasara.

**Supervision:** Ruwan Jayathilaka, Krishantha Wisenthige.

**Validation:** Nehan Perera, Sithesha Samarathunga.

**Writing – original draft:** Kalindu Abeywickrama, Nehan Perera, Sithesha Samarathunga, Harshani Pabasara, Ruwan Jayathilaka, Krishantha Wisenthige.

**Writing – review & editing:** Ruwan Jayathilaka.

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
