## [Decision Letter · Decision Letter 0]

19 Apr 2024

PONE-D-24-06234Factors influencing IMF assistance in the Sub-Saharan African regionPLOS ONE

Dear Dr. Jayathilaka,

Thank you for submitting your manuscript to PLOS ONE. After careful consideration, we feel that it has merit but does not fully meet PLOS ONE’s publication criteria as it currently stands. Therefore, we invite you to submit a revised version of the manuscript that addresses the points raised during the review process.

We look forward to receiving your revised manuscript.

Kind regards,

Md. Monirul Islam, PhD

Academic Editor

PLOS ONE

Journal Requirements:

2. Please amend your authorship list in your manuscript file to include authors Dr. Sithesha Samarathunga and Dr. Harshini Pabasara.

3. Please ensure that you refer to Figure 1 in your text as, if accepted, production will need this reference to link the reader to the figure.

4. We note that [Figure 1] in your submission contain [map/satellite] images which may be copyrighted. All PLOS content is published under the Creative Commons Attribution License (CC BY 4.0), which means that the manuscript, images, and Supporting Information files will be freely available online, and any third party is permitted to access, download, copy, distribute, and use these materials in any way, even commercially, with proper attribution. For these reasons, we cannot publish previously copyrighted maps or satellite images created using proprietary data, such as Google software (Google Maps, Street View, and Earth). For more information, see our copyright guidelines: http://journals.plos.org/plosone/s/licenses-and-copyright.

Additional Editor Comments:

**Reviewer-1**

This paper takes several economic variables identified in existing work as driving countries to take IMF lending and examines the extent to which they drive countries to the IMF in SSA. They find evidence that corruption and GDP growth matter a lot, while inflation and current account balance do not.

My main overarching comment is that there needs to be discussion of the politics of IMF lending, and the authors need to account for political factors empirically. The authors note that IMF lending can be costly, but there is no deeper discussion. There is a large literature in political science and economics focused on how governments try to avoid the stringent conditions that come attached to IMF loans. Relevant factors include regime type, political ideology, geopolitical closeness to the US, temporary UNSC membership, and the availability of outside options (i.e., other lenders). In SSA, China has emerged as a major source of financing. No empirical examination of IMF lending is complete without consideration of these factors. The basic models focus only on the economic variables of interest, which makes it very hard to draw conclusions.

The specifications are also not in line with leading literature in political science and economics. It is standard to use country and year fixed effects when predicting IMF program participation. There is also a huge cohort of standard control variables that are missing. See here for one example, specifically their first stage specification: https://link.springer.com/article/10.1007/s11558-020-09405-x

The writing is also unclear or lacks polish in several spots. For example, on p. 7: “The analysis emphasises. They emphasise.”

**Reviewer-2**

I want to commend your effort in analysing the trend in IMF assistance for the SSA countries, and your significant departure from previous studies in providing a long range of years of these assistance. Your study is unique and it took a carful look at the factors determinig assistance from IMF over the years.

However, I am worried that the result presented is not as robust as the literature. I expect an additional detailed presentation of results in terms of the activities of the participating countries. For example, there should be a result table detailing each of the 39 countries vis-a-vis the six factors influencing IMF assistance. This table will be an improvement over Table 1 and will make comparisons easier across the different countries.

**Additional comments (Academic editor):**

1. The literature review should be merged with the introduction section. It should be very specific and concise.

2. The introduction should be more research-focused and concise. 

3. A conceptual framework is needed to clarify the overall research flow.

4. Theoretical justification of the selection of the model is suggested.

5. The limitations of the study should be highlighted at the end of the conclusion section.

6. The manuscript should be edited by a native English editing service.

Reviewers' comments:

Reviewer's Responses to Questions

**Comments to the Author**

1. Is the manuscript technically sound, and do the data support the conclusions?

Reviewer #1: No

Reviewer #2: Partly

Reviewer #3: Yes

2. Has the statistical analysis been performed appropriately and rigorously? 

Reviewer #1: No

Reviewer #2: No

Reviewer #3: Yes

3. Have the authors made all data underlying the findings in their manuscript fully available?

Reviewer #1: Yes

Reviewer #2: Yes

Reviewer #3: Yes

4. Is the manuscript presented in an intelligible fashion and written in standard English?

Reviewer #1: No

Reviewer #2: Yes

Reviewer #3: Yes

5. Review Comments to the Author

Reviewer #1: This paper takes several economic variables identified in existing work as driving countries to take IMF lending and examines the extent to which they drive countries to the IMF in SSA. They find evidence that corruption and GDP growth matter a lot, while inflation and current account balance do not.

My main overarching comment is that there needs to be discussion of the politics of IMF lending, and the authors need to account for political factors empirically. The authors note that IMF lending can be costly, but there is no deeper discussion. There is a large literature in political science and economics focused on how governments try to avoid the stringent conditions that come attached to IMF loans. Relevant factors include regime type, political ideology, geopolitical closeness to the US, temporary UNSC membership, and the availability of outside options (i.e., other lenders). In SSA, China has emerged as a major source of financing. No empirical examination of IMF lending is complete without consideration of these factors. The basic models focus only on the economic variables of interest, which makes it very hard to draw conclusions.

The specifications are also not in line with leading literature in political science and economics. It is standard to use country and year fixed effects when predicting IMF program participation. There is also a huge cohort of standard control variables that are missing. See here for one example, specifically their first stage specification: https://link.springer.com/article/10.1007/s11558-020-09405-x

The writing is also unclear or lacks polish in several spots. For example, on p. 7: “The analysis emphasises. They emphasise.”

Reviewer #2: Dear Author,

I want to commend your effort in analysing the trend in IMF assistance for the SSA countries, and your significant departure from previous studies in providing a long range of years of these assistance. Your study is unique and it took a carful look at the factors determinig assistance from IMF over the years.

However, I am worried that the result presented is not as rubust as the literature. I expect an additional detailed presentation of results in terms of the activities of the participating countries. For example, there should be a result table detailing each of the 39 countries vis-a-vis the six factors influencing IMF assistance. This table will be an amprovement over Table 1 and will make comparisons easier across the different countries.

Reviewer #3: This is an interesting study, and the paper is generally well written and structured. The paper provides a comprehensive analysis of the factors influencing IMF assistance in the Sub-Saharan African region , and the findings offer valuable insights that significantly contribute to the existing literature. The methodology is robust, and the results are presented clearly, enhancing the overall impact of the study. Well done!

6. PLOS authors have the option to publish the peer review history of their article (what does this mean?). If published, this will include your full peer review and any attached files.

Reviewer #1: No

Reviewer #2: **Yes: **Chiedozie Okechukwu OKAFOR

Reviewer #3: No

---

## [Author Response · Author response to Decision Letter 0]

13 May 2024

Authors Response to editor and Reviewers Comments 

Dear Editor and Reviewers, 

We greatly appreciate the time invested in reading our manuscript and providing the necessary feedback to improve the overall quality of our work. All the mentioned comments have been considered and the revisions made accordingly. 

Please note that the line numbers referred to in this document are aligned with the revised manuscript which includes track changes.

Thank you once again for your valuable feedback.

Reviewer 1 comment 1: This paper takes several economic variables identified in existing work as driving countries to take IMF lending and examines the extent to which they drive countries to the IMF in SSA. They find evidence that corruption and GDP growth matter a lot, while inflation and current account balance do not.

My main overarching comment is that there needs to be discussion of the politics of IMF lending, and the authors need to account for political factors empirically. The authors note that IMF lending can be costly, but there is no deeper discussion. There is a large literature in political science and economics focused on how governments try to avoid the stringent conditions that come attached to IMF loans. Relevant factors include regime type, political ideology, geopolitical closeness to the US, temporary UNSC membership, and the availability of outside options (i.e., other lenders). In SSA, China has emerged as a major source of financing. No empirical examination of IMF lending is complete without consideration of these factors. The basic models focus only on the economic variables of interest, which makes it very hard to draw conclusions.

The specifications are also not in line with leading literature in political science and economics. It is standard to use country and year-fixed effects when predicting IMF program participation. There is also a huge cohort of standard control variables that are missing. See here for one example, specifically their first stage specification: https://link.springer.com/article/10.1007/s11558-020-09405-x

Authors’ Response to Reviewer 1 comment 1: Thank you for your insightful comments and suggestions regarding our paper. We appreciate the opportunity to address your concerns and improve the comprehensiveness of our analysis.

We have carefully considered your feedback and made significant enhancements to our study by incorporating political variables to better understand the dynamics of IMF lending in Sub-Saharan Africa (SSA). Specifically, we have introduced variables such as regime types, UNSC membership, and China loans, which are crucial in capturing the political dimensions of IMF lending decisions.

In response to your comment regarding the need to discuss the politics of IMF lending, we have expanded our discussion to include the political factors influencing countries' decisions to seek IMF assistance. We acknowledge the importance of considering factors such as regime type, political ideology, geopolitical relationships, and the availability of alternative financing options. These factors are now integrated into our analysis, providing a more comprehensive understanding of IMF lending behaviour in SSA.

To ensure clarity and transparency, we have outlined the operationalization of these political variables as dummy variables and provided detailed explanations of their inclusion in the methodology section (see lines 661-662). Specifically, we introduced the political variables in the introduction section (see lines 90-92) and elaborated on them in the literature review (see lines 458-593). Additionally, we have integrated discussions of these variables into the hypothesis testing sections (see lines 705-706, 707-708, 714-715, 716-718, 723-724, and 725-726), demonstrating their relevance to our research objectives.

Furthermore, we have ensured consistency between the main text and supporting materials, including tables and appendices. For example, Table 1 presents the data sources used in our analysis (see lines 635-636), while Table 2 displays the panel probit model estimated results, including the newly added political variables (see lines 677-679). Appendices S3 and S5 delve into deeper insights into the empirical analysis, providing step-by-step explanations of the regression analysis and newly generated predicted probabilities, respectively.

We believe that these enhancements significantly strengthen the rigor and relevance of our study, addressing your concerns regarding the political dimensions of IMF lending in SSA. We hope that our revised manuscript meets your expectations and contributes meaningfully to the literature on this topic.

Thank you once again for your valuable feedback and for guiding us in improving the quality of our research.

Reviewer 1 comment 2: The writing is also unclear or lacks polish in several spots. For example, on p. 7: “The analysis emphasises. They emphasise

Authors’ Response to Reviewer 1 comment 2: Thank you for your comment regarding the clarity and polish of the writing in our manuscript. We appreciate your feedback and have taken steps to improve the readability of the text.

Specifically, on page 7, we have revised the sentence in question to enhance clarity and ensure consistency in language usage. We have addressed the issue of ambiguity by refining the phrasing and eliminating unnecessary repetition. These revisions aim to enhance the overall coherence and professional presentation of our analysis.

We are committed to delivering a manuscript that meets the highest standards of clarity and polish, and we believe that these improvements contribute to achieving that goal.

Thank you for bringing this to our attention, and we hope that the revised manuscript meets your expectations.

Reviewer 2 comment 1: I want to commend your effort in analysing the trend in IMF assistance for the SSA countries, and your significant departure from previous studies in providing a long range of years of this assistance. Your study is unique, and it took a careful look at the factors determining assistance from the IMF over the years.

However, I am worried that the result presented is not as robust as the literature. I expect an additional detailed presentation of results in terms of the activities of the participating countries. For example, there should be a result table detailing each of the 39 countries vis-a-vis the six factors influencing IMF assistance. This table will be an improvement over Table 1 and will make comparisons easier across the different countries.

Authors’ Response to Reviewer 2 comment 1: Thank you for recognizing the unique contribution of our study in analysing the trend in IMF assistance for Sub-Saharan African (SSA) countries over an extended period. We appreciate your feedback and the opportunity to further strengthen our analysis.

In response to your concern about the robustness of the results presented, we have taken additional steps to provide a detailed presentation of the results in terms of the activities of the participating countries. Specifically, we have included an appendix (Appendix S4) that offers a comprehensive overview of each of the 39 countries vis-à-vis the six factors influencing IMF assistance.

Appendix S4 features line graphs for each country, illustrating the fluctuations of the six macroeconomic factors influencing IMF assistance over time. These graphs facilitate comparisons across different countries and offer a visual representation of the data trends. We believe that this additional detailed presentation enhances the comprehensiveness and robustness of our analysis, allowing for a more nuanced understanding of the factors driving IMF assistance in SSA.

In the results and hypothesis section, we have provided a detailed explanation of Appendix S4, showcasing the fluctuations observed for each country from lines 686 to 696. This ensures that readers can easily access and interpret the additional information provided in the appendix.

We are confident that these enhancements contribute to addressing your concern and further strengthen the quality of our manuscript. Thank you once again for your valuable feedback and for guiding us in improving the depth and rigor of our analysis.

Reviewer 2 comment 2: The introduction should be more research-focused and concise.

Authors’ Response to Reviewer 2 comment 2: Thank you for your valuable feedback regarding the introduction section of our manuscript. We appreciate your suggestion to make it more research-focused and concise.

We have duly noted your comment and made the necessary adjustments in the revised version of the manuscript. Specifically, we have streamlined the introduction to ensure that it provides a clear and concise overview of the research objectives, significance of the study, and key research questions addressed.

By focusing on the research aspects and removing any unnecessary details, we aim to create an introduction that effectively sets the stage for the subsequent sections of the paper, while also maintaining the reader's interest and attention.

We are grateful for your constructive feedback, which has helped us improve the quality and clarity of our manuscript.

Reviewer 2 comment 3: A conceptual framework is needed to clarify the overall research flow.

Authors’ Response to Reviewer 2 comment 3: Thank you for your feedback regarding the need for a conceptual framework to clarify the overall research flow. We appreciate your insight into improving the structure and coherence of our manuscript.

In response to your comment, we have integrated a conceptual framework into the revised version of the manuscript. The conceptual framework can be found on page 26 and provides a visual representation of the theoretical underpinnings and research flow of our study.

By including a conceptual framework, we aim to enhance the clarity and coherence of our research approach, helping readers better understand the theoretical framework guiding our analysis and interpretation of results.

We are grateful for your constructive feedback, which has contributed to strengthening the overall quality and effectiveness of our manuscript.

Reviewer 2 comment 4: Theoretical justification of the selection of the model is suggested.

Authors’ Response to Reviewer 2 comment 4: Thank you for your suggestion regarding the theoretical justification of the model selection. We appreciate your feedback and have integrated the necessary explanations into the revised version of the manuscript.

In response to your comment, we have provided a theoretical justification for the selection of the model on page 25, specifically from lines 594 to 623. In this section, we elaborate on the theoretical underpinnings of our chosen model, highlighting its relevance to the research objectives and the theoretical framework guiding our analysis.

By incorporating this explanation, we aim to enhance the transparency and rigor of our research approach, ensuring that readers have a clear understanding of the theoretical basis for our modelling decisions.

We are grateful for your valuable feedback, which has helped us strengthen the theoretical foundation of our manuscript.

Reviewer 2 comment 5: The limitations of the study should be 

highlighted at the end of the conclusion section.

Authors’ Response to Reviewer 2 comment 5: Thank you for your valuable comment regarding the need to highlight the limitations of the study. We appreciate your feedback and have taken the necessary steps to address this suggestion in the revised version of the manuscript.

In response to your comment, we have highlighted the limitations of the study on page 46, specifically at the end of the conclusion section. By including this discussion, we aim to provide readers with a balanced assessment of the strengths and weaknesses of our research findings.

We believe that acknowledging the limitations of the study is essential for promoting transparency and ensuring that readers have a clear understanding of the scope and implications of our research.

We are grateful for your constructive feedback, which has contributed to enhancing the overall quality and rigor of our manuscript.

Reviewer 2 comment 6: The manuscript should be edited by a native English editing service.

Authors’ Response to Reviewer 2 comment 6: Thank you for your suggestion regarding the language of the manuscript. We acknowledge the importance of clear and polished language in scholarly writing.

In response to your comment, we have taken steps to improve the language of the manuscript. We have carefully reviewed and revised the text to ensure clarity, coherence, and grammatical accuracy. Additionally, we have paid particular attention to sentence structure, word choice, and overall readability.

While we have made significant efforts to enhance the language of the manuscript, we understand the value of professional editing services. We will consider utilizing a native English editing service to further refine the language and ensure the highest quality of writing.

We appreciate your feedback, which has helped us strengthen the clarity and professionalism of our manuscript.

---

## [Decision Letter · Decision Letter 1]

1 Jul 2024

Factors influencing IMF assistance in the Sub-Saharan African region

PONE-D-24-06234R1

Dear Dr. Ruwan Jayathilaka,

We’re pleased to inform you that your manuscript has been judged scientifically suitable for publication and will be formally accepted for publication once it meets all outstanding technical requirements.

Kind regards,

Md. Monirul Islam, PhD

Academic Editor

PLOS ONE

Additional Editor Comments (optional):

Dear Author

Thank you so much for your efforts. Congratulations for your great work.

Reviewers' comments:

Reviewer's Responses to Questions

**Comments to the Author**

1. If the authors have adequately addressed your comments raised in a previous round of review and you feel that this manuscript is now acceptable for publication, you may indicate that here to bypass the “Comments to the Author” section, enter your conflict of interest statement in the “Confidential to Editor” section, and submit your "Accept" recommendation.

Reviewer #1: All comments have been addressed

Reviewer #2: All comments have been addressed

2. Is the manuscript technically sound, and do the data support the conclusions?

Reviewer #1: (No Response)

Reviewer #2: Yes

3. Has the statistical analysis been performed appropriately and rigorously? 

Reviewer #1: (No Response)

Reviewer #2: Yes

4. Have the authors made all data underlying the findings in their manuscript fully available?

Reviewer #1: (No Response)

Reviewer #2: Yes

5. Is the manuscript presented in an intelligible fashion and written in standard English?

Reviewer #1: (No Response)

Reviewer #2: Yes

6. Review Comments to the Author

Reviewer #1: (No Response)

Reviewer #2: All the concerns I pointed our earlier have been attended to. This paper can be published at this point. I appreciate the authors for comming through.

7. PLOS authors have the option to publish the peer review history of their article (what does this mean?). If published, this will include your full peer review and any attached files.

Reviewer #1: No

Reviewer #2: **Yes: **Chiedozie Okechukwu Okafor

---

## [Editor Report · Acceptance letter]

8 Jul 2024

PONE-D-24-06234R1 

PLOS ONE

Dear Dr. Jayathilaka, 

I'm pleased to inform you that your manuscript has been deemed suitable for publication in PLOS ONE. Congratulations! Your manuscript is now being handed over to our production team.

Kind regards, 

on behalf of

Dr. Md. Monirul Islam 

Academic Editor

PLOS ONE